# Nanotechnology in Advancing Chimeric Antigen Receptor T Cell Therapy for Cancer Treatment

**DOI:** 10.3390/pharmaceutics16091228

**Published:** 2024-09-20

**Authors:** Xuejia Kang, Nur Mita, Lang Zhou, Siqi Wu, Zongliang Yue, R. Jayachandra Babu, Pengyu Chen

**Affiliations:** 1Materials Research and Education Center, Materials Engineering, Department of Mechanical Engineering, Auburn University, Auburn, AL 36849, USA; lzz0028@auburn.edu (L.Z.); szw0165@auburn.edu (S.W.); 2Department of Drug Discovery and Development, Harrison College of Pharmacy, Auburn University, Auburn, AL 36849, USA; nzm0076@auburn.edu (N.M.); zzy0065@auburn.edu (Z.Y.); ramapjb@auburn.edu (R.J.B.); 3Faculty of Pharmacy, Mulawarman University, Samarinda 75119, Kalimantan Timur, Indonesia

**Keywords:** CAR-T therapy, nanotechnology, barriers, central nervous system (CNS) tumors

## Abstract

Chimeric antigen receptor (CAR) T cell therapy has emerged as a groundbreaking treatment for hematological cancers, yet it faces significant hurdles, particularly regarding its efficacy in solid tumors and concerning associated adverse effects. This review provides a comprehensive analysis of the advancements and ongoing challenges in CAR-T therapy. We highlight the transformative potential of nanotechnology in enhancing CAR-T therapy by improving targeting precision, modulating the immune-suppressive tumor microenvironment, and overcoming physical barriers. Nanotechnology facilitates efficient CAR gene delivery into T cells, boosting transfection efficiency and potentially reducing therapy costs. Moreover, nanotechnology offers innovative solutions to mitigate cytokine release syndrome (CRS) and immune effector cell-associated neurotoxicity syndrome (ICANS). Cutting-edge nanotechnology platforms for real-time monitoring of CAR-T cell activity and cytokine release are also discussed. By integrating these advancements, we aim to provide valuable insights and pave the way for the next generation of CAR-T cell therapies to overcome current limitations and enhance therapeutic outcomes.

## 1. CAR-Based Therapy in Cancer

Cancer, a step-by-step disease, remains one of the most challenging to treat. Conventional therapies often fall short due to issues such as drug resistance, lack of lasting immune memory, harmful effects on normal organs, and the complex nature of the tumor microenvironment [1,2]. CAR-cell therapy has emerged as a promising avenue for treating various cancer malignancies [3]. Notable CAR-T cell products, including axicabtagene ciloleucel (Yescarta) and tisagenlecleucel (Kymriah), have shown remarkable efficacy in patients with hematological cancers [4]. Recently, new products, such as ciltacabtagene autoleucel (Carvykti), idecabtagene vicleucel (Abecma), brexucabtagene autoleucel (Tecartus), and lisocabtagene maraleucel (Breyanzi) have received FDA approval, expanding the therapeutic arsenal for blood cancers. Besides CAR-T, CAR-macrophage and CAR-NK cell therapy have also been developed. And over the past few decades, nanotechnology has been offering significant benefits for CAR-based cell therapy [5,6]. Firstly, nanotechnology can enhance the design and construction of CARs by enabling more precise and efficient engineering of the receptors [7]. Furthermore, nanomaterials can be used to improve the stability and functionality of CARs, ensuring they bind more effectively to target antigens on cancer cells [8]. A notable example of gold nanoparticles (Au NPs), with their non-immunogenic properties and ability to be eliminated through glomerular filtration, hold potential as effective and non-toxic tools in CAR-T cell construction [9], leading to improved specificity and reduced off-target effects for safer and more effective CAR-T cell therapy. Secondly, nanoparticles play a role in the transfection of T cells, which is crucial for the success of CAR-T cell therapy employed to deliver genetic material encoding the CAR into T cells with high efficiency and minimal toxicity [10]. These nanocarriers protect the genetic material during delivery and facilitate its entry into T cells, ensuring robust CAR expression and enhancing the overall therapeutic efficacy. Thirdly, expanding CAR-T cells to sufficient numbers is a critical preparatory step for therapeutic use. Nanotechnology assists in the development of advanced culture systems that provide optimal environments for T-cell proliferation [11,12]. Nanomaterials can inhibit immunosuppressive factors in the cellular microenvironment while promoting the immune active elements, facilitating more efficient and scalable CAR-T cell expansion [13]. Fourthly, nanotechnology enables precise modifications to be made to CAR-T cells, enhancing their functionality and persistence in the body [14]. For instance, nanoparticles can be used to deliver additional agents to CAR-T cells, helping them cross barriers, evade clearance, and persist in the hostile tumor microenvironment [11]. These modifications can improve the durability and effectiveness of CAR-T cell therapy. Fifthly, real-time monitoring of CAR-T cell function is essential for assessing the therapeutic progress and the success of the treatment. Nanotechnology offers innovative solutions for non-invasive, precise monitoring. For instance, due to the unique optical property of AuNPs in biomedical imaging [15], they can be engineered to act as imaging agents or biosensors to track the location, proliferation, and activity of CAR-T cells within the body. This enables timely adjustments to be made to the treatment regimen, ensuring optimal therapeutic outcomes (Figure 1).

Bio-nanotechnology also allows for the control of side effects of CAR therapy: large molecular-weight antibodies like tocilizumab are significant in minimizing the side effects of CAR therapy but challenges remain for their delivery, due to their size and complexity. Strategies such as the use of hyaluronate–gold nanoparticles, as seen in tocilizumab formulations, and infliximab–modified gold nanorods, have shown promise [16,17]. Another promising strategy involves the use of antibody-mimicking peptides, which are around 50 amino acids in length and can be engineered to replicate the binding specificity and affinity of the original antibodies while being significantly smaller in size [18]. Notably, four phage mimics YHTTDKLFYMMR, YSAYEFEYILSS, KTMSAEEFDNWL, and LTSHTYRSQADT) have been shown to mimic the activity of tocilizumab [19]. Furthermore, utilizing stimuli-responsive nanovehicles that release their cargo in response to specific environmental triggers, such as pH changes or the presence of certain enzymes in the tumor microenvironment [20], improve the circulation time of the antibodies in the bloodstream ensuring that the therapeutic agents reach their intended site of action with minimal off-target effects [21]. Nanodynamic therapies (NDTs) can activate anti-tumor immunity, thereby enhancing the efficacy of immunotherapy [22]. A notable example is photodynamic therapy, which utilizes nanoparticles to enhance the anti-tumor activity of CAR-T cells [23].

Moreover, the potential for nanotechnology to be integrated with multitarget molecular therapies opens new avenues [18,24]. CARs are engineered to redirect T lymphocytes and other immune cells towards specific antigens [25]. However, tumor-associated macrophages (TAMs), myeloid-derived suppressor cells (MDSCs), and cancer-associated fibroblasts (CAFs) can impair CAR-T cell efficacy by producing inhibitory substances that affect the therapy’s effectiveness [26,27]. To address the challenges in CAR-T cell therapy, nanoparticles with multimolecular targets can be engineered to target those cells within the tumor microenvironment [28,29]. For instance, combining all-trans retinoic acid (ATRA) with CAR-T cell therapy can mitigate the suppressive effects of myeloid-derived suppressor cells (MDSCs) [30]. Moreover, the decrease in TGF-b and IL10 using nanoparticle also benefits the efficacy of CAR-T [31]. The combination of CAR-T with chemotherapy delivered by NPs is able to inhibit the immunosuppressive cells including T reg and MDSC [32,33]. The delivery of chemotherapy with nanoparticles can also reduce the burden of tumors and increase the damage associated with pattern release, which further potentially affects the effect of CAR-T [32,34].

### State-of-the-Art CAR Therapy for Brain Tumors

Brain tumors represent one of the most formidable challenges in oncology, with an overall incidence of about 4–5 per 100.000 individuals [35]. Gliomas, particularly glioblastomas, are the most common form of primary brain tumors [36,37], characterized by rapid growth, extensive infiltration into surrounding brain tissue, and a high propensity for recurrence. These malignant brain tumors, originating from glial cells, notably astrocytes, primarily affect adults aged 50 to 70 but can occur at any point across the lifespan [36,37]. The current standard treatment options include surgery, followed by radiotherapy and chemotherapy with temozolomide (TMZ) [35]. Despite these treatments, glioblastomas often recur, underscoring the need for more effective therapies. CAR-T cell therapy holds significant promise for treating brain tumors. Pre-clinical efforts are increasingly focused on extending CAR-T cell therapy to solid tumors, including brain tumors. Identified targets for these tumors include EGFR variant III (EGFRvIII), interleukin–13 receptor subunit alpha-2 (IL13Rα2), and HER2 [38,39]. Innovative strategies such as dual-targeting CAR-T cells, exemplified by TanCAR (targeting IL–13Rα2 and ephrin type-A receptor 2 (EphA2)), and modifications like CXCR2-engineered CAR-T cells, are being developed to enhance efficacy against solid tumors [40,41]. Additionally, CAR-T cells targeting the tumor microenvironment, such as FAP-CAR-T cells, show promise in modulating cancer-associated fibroblasts in lung and pancreatic cancers [42]. Targeting endothelial cell factors, like vascular endothelial growth factor receptor 2 (VEGFR2), also enhances CAR-T cell penetration and anti-tumor effects [43]. The complexity and high costs associated with CAR-T cell modifications necessitate more cost-effective approaches, such as combinatorial therapies [42]. The recent CAR-T cell products specifically targeting brain tumors are summarized below (Table 1).

## 2. Current Challenges of CAR-Based Therapy

### 2.1. Challenges in Composing CAR-Based Therapy and Advantages of Nano-Vector

Current FDA-approved CAR-T primarily utilize lentivirus and/or retrovirus for CAR gene transduction [57,58]. Using viral vectors ensures high transfection efficacy and long-term gene expression, although transduction efficiency can vary, impacting therapy consistency [59]. Additionally, composing CAR cell therapy using viral vectors introduces higher variability from batch to batch and storage concerns [60]. Worse still, using viral vectors for CAR-T therapy may pose a risk of insertional mutagenesis, potentially leading to further malignancies [61]. Immunogenicity, where viral proteins trigger immune responses that may destroy modified T cells, is also a significant concern [61]. Moreover, chemical modification and ligand attachment are challenging with viral vectors, and the manufacturing process is complex and costly, necessitating strict safety protocols to avoid contamination [62]. Viral vectors also have limited packaging capacity, restricting the size of CAR constructs [61]. These issues highlight the need for ongoing research to improve the safety and efficacy of viral vector-based therapies. To overcome these challenges, non-viral physical methods have gained attention. Electroporation, in particular, has shown promise by creating temporary pores in the cell membrane through the application of an electrical field, allowing for more efficient gene transfer [63,64]. 

Microinjection, though less commonly used in large-scale transduction due to its single-cell approach, remains valuable for immunotherapy [65]. Sonication, a newer method, uses ultrasonic waves to enhance the permeability of cell membranes, facilitating gene uptake [66]. These physical techniques not only improve transfection efficiency but also reduce reliance on viral vectors [67]. However, these methods can also cause damage to cells, presenting a significant hurdle in their application.

Compared to viral vector and physical methods, nanotechnology offers a promising alternative for advancing CAR-based therapy [68]. Importantly, since nanomaterials are easier to manufacture, store, and transport, nanotechnology can significantly reduce the cost of CAR-based therapy. In addition, NPs can serve as versatile delivery platforms for CAR genes and other therapeutic agents (Figure 2). The workflow for generating NPs as carriers for CAR cargo involves isolating and activating T cells, NP transduction and optimizing the formulation,, infusing the NP-based products into the patient, and achieving in vivo expression of CAR [7]. Yu et al. [28] described the in vitro engineering of CAR-T cells using self-assembled nanoparticles (SNPs) designed to minimize toxicity [69]. These SNPs were created by encapsulating plasmid DNA (pDNA) within a matrix of polyethylene glycol (PEG) and cationic polymers, including polyamidoamine (PAMAM) and PEI [69]. Lipid nanoparticles (LNPs) are being used more frequently to target antigen-presenting cells (APCs) in order to stimulate T-cell responses in preclinical in vivo models [70]. Modified cationic lipids have been shown to trigger strong CD8+ and CD4+ T-cell responses [71]; PEGylated nanoparticles offer a biocompatible system for gene delivery, improving both the circulation time and stability of vaccines in vivo [72]. PEG-modified nanoparticles are good examples of providing a biocompatible platform for gene transfer. This NP enhanced the stability of vaccines in vivo [73]. In addition, LNP can encapsulate antigens and deliver tumor antigen directly to APCs, such as dendritic cells (DCs) [74]. For instance, empty LNP (eLNP) induced maturation of DCs [75]. Furthermore, the administration of eLNP resulted in the upregulation of CD40 and the induction of cytokine production, exhibiting an age-dependent response [75]. Due to their lipid composition, dendritic cells (DCs) preferentially uptake LNPs [76,77]. Furthermore, surface modifications of these nanoparticles can enhance their uptake by DCs, leading to improved antigen presentation and more effective T-cell activation [78].

LNPs are also used to deliver mRNA encoding tumor antigens or immune-modulatory proteins to APCs [79]. Moreover, the incorporation of other ingredients such as C–24 alkyl phytosterols can improve the gene transfection efficacy [79]. Rurik et al. [80] found coupling a CD5 antibody to an LNP to form CD5/LNP can be used to deliver CAR mRNA and targeting fibroblast-activating protein (FAP) can minimize cardiac injury [80]. The LNP vector successfully targeted splenic T cells in mice with high CD5 expression, recognizing cardiac fibroblasts, ultimately reducing fibrosis and improving cardiac function [80]. Furthermore, combining antigens with immune-stimulatory adjuvants in LNPs can further enhance T-cell activation [81]. Adjuvants such as Toll-like receptor (TLR) agonists or other immune modulators can be incorporated into LNPs to amplify the overall immune response [82]. For instance, MHC class I antigenic-peptide ligand encapsulated in an LNP resulted in increased T cell expansion in vivo [82]. Additionally, LNP-based therapies have been developed for RNA interference (RNAi) to treat transthyretin amyloidosis-induced polyneuropathy, ushering in the LNP-RNA era [83,84]. Billingsley et al. [30] reported that LNPs can deliver mRNA to primary human T cells with much lower toxicity than traditional electroporation methods. This was achieved by combining a set of twenty-four ionizable lipids with cholesterol, phospholipids, and lipid-anchored PEG, then mixing them with CAR mRNA using a microfluidic device to form LNPs [85].

### 2.2. Challenges for the Application of CAR-Based Therapy

Compared to hematological cancers, the application of CAR-T cell therapy in solid tumors faces more challenges and solid tumors are inherently less responsive to CAR-T cell therapy [86]. Firstly, CAR-T cells encounter substantial difficulty in accessing solid tumors due to physical barriers [87,88]. In the case of brain tumors, the blood–brain barrier represents a particularly restrictive obstacle that hinders the delivery of CAR-T cells to the brain [87]. Furthermore, the tumor immunosuppressive microenvironment, which includes cancer-associated fibroblasts [42], tumor-associated macrophages [89], and endothelial cells [43], poses additional challenges for CAR-T cell penetration [90]. These cells secrete immune-suppressive factors that further impede CAR-T cell function and contribute to the acceleration of their exhaustion [90]. Moreover, the immune-suppressive factors within the tumor microenvironment exacerbate the recruitment of harmful cells [90]. Furthermore, antigen heterogeneity in solid tumors complicates the efficacy of CAR-T cells, as tumor-associated antigens (TAAs) are also expressed in normal cells in other regions, causing tumor off-target effects [91]. This is particularly perilous in the brain, where off-tumor targeting of normal cells can result in severe toxicity, as observed with MAGE-A3 TCR-engineered T cells [92]. Additionally, solid tumors tend to form metastatic sites with different TAAs, presenting further challenges for CAR-T cell efficacy [91]. Hypoxic conditions and metabolic competition within solid tumors also hinder CAR-T cell function, as hypoxia-driven metabolic changes in both tumor and immune cells contribute to the development of an immunosuppressive tumor microenvironment (TME) [93]. 

As stated, solid TMEs are inherently immunosuppressive [94]. Anatomical features in brain tumors limit T cell infiltration, and intrinsic factors in glioma, based on their mutational profile and gene expression patterns, result in a more suppressive TME [94]. A study by Kohanbash and colleagues demonstrated that mutations in isocitrate dehydrogenase genes in glioma cells suppress STAT1 expression, leading to reduced accumulation of CD8 + T cells, type 1-associated effector molecules, and chemokines such as CXCL10, thereby forming the tumor immunosuppressive environment [95,96]. Nanotechnology offers solutions to address TME by creating nanocarriers that deliver carefully selected combinations of immunomodulatory drugs, minimizing adverse effects [97]. Immune responses such as PD–1 on T cells and PD-L1 in the TME contribute to CAR-T cell exhaustion. The use of PD–1/PD-L1 antagonists can modify the immunosuppressive TME and prolong CAR-T cell persistence [98,99]. Magnetic nanoclusters encapsulated with anti-PD–1 antibodies can also modify the TME [100]. Additionally, cell membrane-coated magnetosomes containing anti-CD28 and MHC-I can reshape the TME.

Solid tumors are characterized by a highly hypoxic TME [101,102]. Central nervous system-specific anatomical characteristics, including low oxygen and high pressure, further compromise long-term CAR-T cell persistence [103]. Increased hypoxia-inducible factor–1 alpha (HIF–1α) activity and hypoxia in tumor tissues have been correlated with poor prognosis in cancer patients [104]. Hypoxia has been shown to upregulate PD-L1 expression by tumor cells and promote tumor proliferation [105]. NPs can be designed to release their cargo in response to the hypoxic conditions often found in solid tumors, ensuring that CAR-T cells or supportive agents are delivered where they are most needed [106,107]. Moreover, designed lipid nanoparticles coated with iRGD (a tumor-targeting peptide) and loaded with a PI3K inhibitor (targeting immunosuppressive tumor cells) and α-GalCer (a T cell stimulant) inhibit suppressor cells within the solid tumor environment and simultaneously stimulate T cells [97]. In addition, due to the homing effects, the CD–45-modified liposomes benefit T cell infiltration in B16F10 melanoma tumors [108]. Nanoparticles can be used to deliver cytokines or other modulatory agents in a controlled manner, enhancing CAR-T cell function and persistence while minimizing systemic toxicity [108]. Some cytokine inhibitors, such as TGF-β1 antagonists, are loaded into NPs and used to modulate the TME [109]. These inhibitors benefit the activation of CD8 + T cells but can cause cardiac toxicity and autoimmune pathology when used systemically [110]. Cytokines such as IL2 and IFN-r benefit from the increased efficacy of CAR-T therapy [111,112]. NPs can be designed to provide sustained stimulation to CAR-T cells, promoting their expansion and persistence in the body. These NPs can mimic the signals provided by antigen-presenting cells, further enhancing CAR-T cell activation and proliferation. 

## 3. Nanotechnology in Enhancing the Efficacy of CAR Cell Therapy

Nanotechnology can significantly enhance the antigen recognition capabilities of CAR-based therapies in solid tumors by utilizing nanoparticle-based artificial antigen-presenting cells [113]. Carbon nanotubes, for instance, improved activated T cells due to their high surface-area-to-volume ratio [114]. These nanotubes can bind more antigens, thereby increasing the presentation efficiency of anti-CD3 antibodies to T cells [115]. Moreover, carbo-nanotubes require less IL2 to achieve the same level of T cell expansion [116].

The precise spatial organization of CAR components to help CAR products overcome barriers is crucial to improve the efficacy of CAR-T cell therapy and overcome the various barriers [88]. Nanotechnology can enhance the ability of CAR-T cells to bypass obstacles or destroy physical and biochemical barriers, thereby improving their functionality [108,117]. For example, nanogels or liposomes can facilitate the delivery of CAR-T cells to tumor sites, enhancing targeting precision and minimizing off-tumor effects [108,117]. Liposomes containing A2A receptor-specific antagonists can be attached to CAR-T cells to circumvent the immunosuppressive effects of adenosine in the TME, thus enhancing CAR-T cell functionality [118]. Additionally, protein nanogel loaded with IL–15 superagonist (IL–15sa) and an antibody against CD45 can benefit the activation of CAR-T cells [119]. The above combination can enhance CAR-T cell proliferation, persistence, and activity by providing additional stimulatory signals and reducing inhibitory signals within the TME [119].

For brain tumors, the BBB is a major obstacle in delivering therapeutic agents to reach the brain [120] The size and surface charge of therapeutics are critical for BBB penetration. Furthermore, brain tumors are often surrounded by a dense extracellular matrix, which impedes CAR-T migration to tumor regions. To overcome the aforementioned challenges, researchers are paying particular attention. For instance, Zhao and colleagues synthesized a tumor-targeting NP, HA@Cu_2−x_S-PEG (PHCN), which can degrade the tumor extracellular matrix by converting light energy into local heat [120]. Additionally, for brain tumors, nanoparticles can be modified to cross the BBB using various strategies, including surface modifications with ligands that bind to receptors on BBB endothelial cells. Ligands such as transferrin, lactoferrin, and certain peptides (e.g., TAT peptide) can facilitate receptor-mediated transcytosis across the BBB [121,122,123]. These modifications allow for the direct delivery of therapeutic agents to brain tumors, potentially improving treatment effectiveness [121], although further studies are needed in this area.

Furthermore, NPs can be functionalized with ligands or antibodies that recognize tumor-specific antigens, ensuring that CAR-T cells are activated only in the presence of a tumor [97,124]. And techniques such as laser or magnetic effects can also enhance CAR-T cell efficacy [125]; however, lasers pose a risk of inducing damage to human skin and require further research [125].

Reducing tumor burden is also a rational strategy to complement many immunotherapies, including CAR-T therapy. NPs containing cytotoxic agents can reduce tumor burden, potentially lowering the required dose of CAR T-cells [126,127]. In addition, some of these therapeutic agents can induce immunogenic cell death, further increasing the efficacy of CAR-T [126,127]. However, it is essential to avoid impairing CAR-T cell function during the combinational process.

NPs can also be employed to deliver agents that modulate the extracellular microenvironment, thereby enhancing the efficacy of CAR-T cell therapy. When designed with sizes between 50 and 150 nm and utilizing cationic cycle-based NPs and no spherical architecture, they can thereby protect the CAR from the extracellular environment [128,129]. Polymeric NPs can be designed to exploit endogenous transport mechanisms, such as adsorptive-mediated endocytosis or carrier-mediated transport, to cross the BBB [120]. Thus, these NPs can be loaded with CAR-T cells or drugs to enhance CAR-T cell activity (Figure 3).

## 4. Nanotechnology in Mitigating the Adverse Effects of CAR-T Cell Therapy

### 4.1. Adverse Events (AEs) in CAR-T Therapy

Adverse events (AEs) following CAR-T therapy primarily include cytokine release syndrome (CRS), hemophagocytic lymphohistiocytosis/macrophage activation syndrome (HLH/MAS), on-target, off-tumor effects, and neurologic toxicity. Symptoms such as high fever [130], malaise, fatigue, myalgia, nausea, anorexia, [131,132], tachycardia, hypotension, [133], and multi-organ dysfunctions are driven by these mechanisms [134,135]. Neurologic toxicity, presenting as headaches, tremors, and dysgraphia, arises from elevated cytokines and T cells in the cerebrospinal fluid [136]. Additionally, on-target, off-tumor toxicity can cause organ failure due to shared antigens in normal tissues [133]. Infections and bacterial contamination also pose significant challenges to the biosafety of CAR-T cell therapy. 

CAR-T-induced cytokine release can disrupt the integrity of the BBB, leading to neurological complications. Elevated cytokines, including IL–6 [137,138], IFN-γ [139,140], and TNF-α [141], play pivotal roles in disrupting the BBB-endothelial cells [142,143], facilitating the expression of tight junction markers and the entry of excessive cytokines into the brain and causing immune effector cell-associated neurotoxicity syndrome (ICANS) [143]. ICANS typically occurs within one to three weeks post-therapy, though delayed onset has also been reported [144]. Up to 67% of patients with leukemia and 62% of patients with lymphoma experience ICANS [144]. Severe ICANS presents with expressive and global aphasia, coma, seizures, obtundation, and stupor [145]. Other symptoms include hyponatremia, hypokalemia, and hypophosphatemia [146,147]. Patients with severe ICANS demonstrate a higher incidence of consumptive coagulopathy [148]. Neurologic toxicity is particularly prevalent in brain tumor patients, including aphasia, tremor, ataxia, hemiparesis, and cranial nerve palsies [149]. Nanotechnology holds great promise in addressing these toxicities and enhancing the safety of CAR-T therapy, as illustrated in Figure 4. 

### 4.2. Nanomedicine for Managing Cytokine-Related Symptoms

General CAR-T cell toxicities include systemic immune activation with CRS being a common manifestation. HLH/MAS, a severe syndrome associated with CRS, involves high ferritin levels and hemophagocytosis [150,151]. Fever is usually the initial symptom of cytokine release syndrome after CAR-T cell infusion, appearing from a few hours after to over a week later [152]. CAR-T also resulted in other adverse events such as malaise, headaches, muscle and joint pains, and anorexia [153]. Mitigating these symptoms is crucial for patient well-being. Tocilizumab, an antibody against IL6, has been approved for CRS treatment [154]. However, its use is often limited by issues such as cytopenia, resultant rheumatoid arthritis, and subsequent infections [130]. Using a nanocarrier loaded with tocilizumab may prevent premature release and avoid these potential risks [155]. Other antagonists, such as siltuximab [156], infliximab [157], and etanercept [158], as well as the IL–1 antagonist anakinra [159], have also been discovered. Nanotechnology, which relies on nanomaterials to shield drugs, shows potential in providing sustained, controlled release and preventing drug degradation [160]. Additionally, modifications in nanomaterials can reduce untargeted toxicity [161]. Therefore, incorporating these anti-CRS agents with nanotechnology can be beneficial (Figure 5).

Refractory CRS often results in pulmonary complications such as edema, hypoxia, dyspnea, and pneumonitis, occasionally necessitating mechanical ventilation [162,163]. However, not all patients are easily accessible for ventilation, and even with ventilation, irreversible damage may occur. Nanomedicine for managing these complications could provide an option. While corticosteroids are widely used for the first-line treatment of CRS-related diseases, patients with refractory CRS often do not respond to them [132,164]. For those with refractory CRS and/or ICANS, alternative third-line anti-cytokine treatment using NPs offers a promising approach. It has been reported that macrophage membrane-coated nanoparticles act as detoxification agents by removing lipopolysaccharide (LPS) [165]. LPS-stimulated macrophage membrane-coated nanoparticles (LMNP) that have been developed exhibit a strong ability to absorb overexpressed IFN-γ and IL–6 in CAR-T treatments [166]. The macrophage membranes used in LMNP contain receptors with a high affinity for proinflammatory cytokines, enabling the management of macrophage overactivation activated by the cytokine loop [166]. As HLH is also associated with cytokines, LMNPs effectively relieve the symptoms of HLH [166]. Given the similarities between CRS in CAR-T therapy and CRS in COVID-19 infections, some drug delivery systems (DDS) may offer insights into alleviating symptoms in CAR-T cell patients. ZnO NPs, approved by the FDA for their safety, may help alleviate CRS and immune responses by restoring homeostasis and releasing anti-inflammatory factors [167,168].

**Figure 5 pharmaceutics-16-01228-f005:**
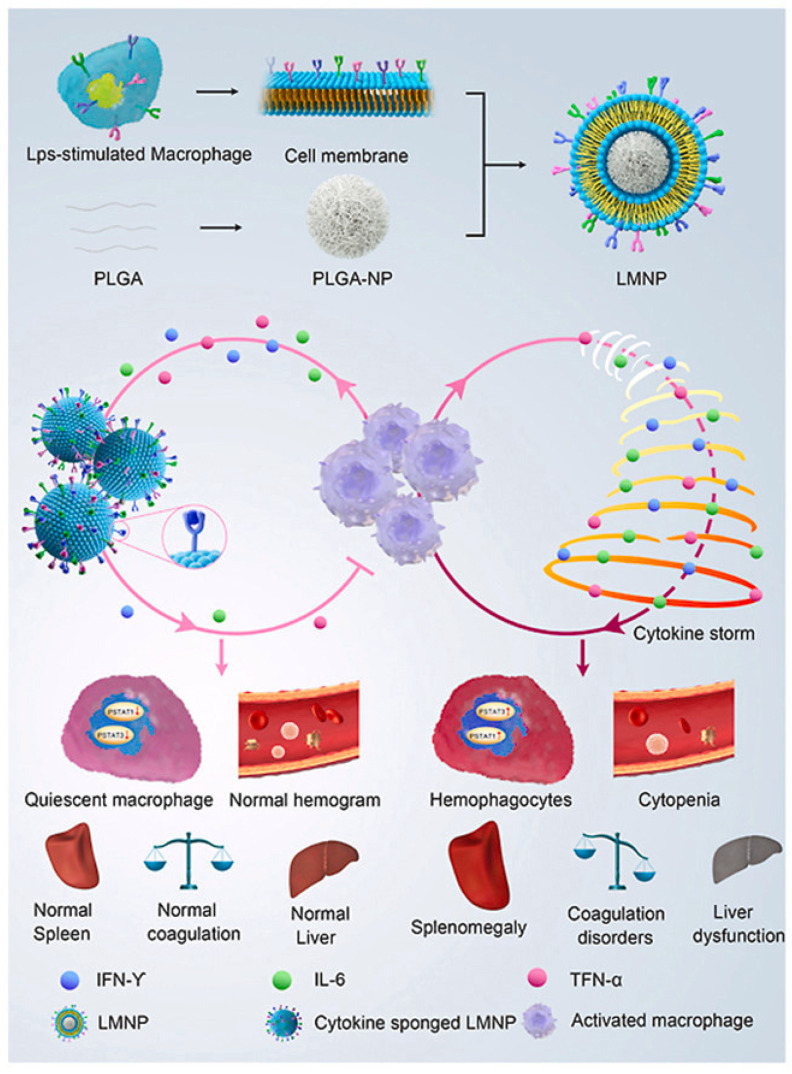
A schematic illustration depicting how cytokine nanosponges (LMNP) suppress overactive macrophages and mitigate systemic cytokine storms, providing treatment for hemophagocytic lymphohistiocytosis (HLH), reprinted with permission [166].

For off-tumor effects, NPs can be modified with specific ligands for tumor cells and mitigate off-tumor effects [169]. Furthermore, NPs can release their payload in response to specific signals, ensuring localized action [170]. Additionally, CAR-T therapy can compromise the immune system, making patients more susceptible to infections [171]. NPs can be engineered to deliver immunomodulators, which help prevent infections and restore immune homeostasis [172]. Nanotechnology-based approaches not only enhance the safety of CAR-T therapy but also improve overall treatment outcomes by maintaining a balanced immune response during therapy [56].

In summary, NPs offer multifaceted benefits in CAR-T cell therapy, including enhanced drug delivery, reduced systemic toxicity, targeted therapy for CRS and ICANS, and prevention of off-target effects and infections. These advancements hold promise for improving the safety and efficacy of CAR-T cell therapies in clinical settings.

## 5. Nanotechnology in Monitoring of CAR Cell Therapy

Nanotechnology-based imaging agents have revolutionized the monitoring and diagnostics of CAR-T therapies, addressing the limitations of traditional methods [24,173]. For instance, classical techniques like enzyme-linked immunosorbent assay (ELISA) face limitations in analysis time, sample size, sensitivity and unsuitability for use in real-time detection [174]. In contrast, bioimmunosensors, a multidisciplinary technology, utilize double-antibody sandwich and competition methods for detection, employing optical and electrical detection mechanisms and providing a more sensitive limit of detection [168]. Innovatively, nanomaterials amplify detection signals by serving as carriers (signal molecules) or catalysts, thereby improving detection sensitivity and overall sensing performance [175,176]. For instance, IL–6 is a significant marker in CRS [110]. A label-free biosensor combined with a AuNP array surface-enhanced Raman spectroscopy (SERS) substrate and a DNA aptamer, achieved high sensitivity and selectivity in IL–6 [177]. Additionally, using NPs as substrate support can capture a large number of target molecules or signal molecules, enhancing detection capabilities [176,178]. Another example is the aptameric dual-channel graphene–Tween 80 field effect transistor (DGTFET), an intelligent biosensing device capable of on-site detection of IFN-γ, TNF-α, and IL–6 within 7 min, with limits of detection (LODs) of 476 × 10^−15^ M for IFN-γ, 608 × 10^−15^ M for TNF-α, and 611 × 10^−15^ M for IL–6 in biofluids [179].

Multiple cytokine detection holds great promise in monitoring CRS. Gao et.al. developed a machine-learning-assisted microfluidic nanoplasmonic digital immunoassay, representing a cutting-edge technology poised for effective cytokine storm monitoring [180]. This innovative approach incorporates high-throughput microarray patterning techniques and utilizes ultrasensitive 100 nm silver nanotubes for signal transduction [181]. As a result, this sensor achieves remarkable sensitivity with LODs for cytokines as follows: 0.91 pg/mL for IL–1β, 0.47 pg/mL for IL–2, 0.46 pg/mL for IL–6, 1.36 pg/mL for IL–10, 0.71 pg/mL for TNF-α, and 1.08 pg/mL (Figure 6A) [181]. This technology not only shows great potential for managing COVID-19 but also holds promise for adapting to CAR-T cell therapy. Based on this, the technique was further developed to monitor CAR-T cell activity and adverse events using a microfluidic leukemia-on-a-chip model [181]. The advanced platform demonstrated that CAR-T cells effectively eliminated B-ALL cells in a dose-dependent manner. In this work, AuNPs produced a strong nanoplasmonic scattering signal, enabling the successful monitoring of increasing cytokine levels over time (Figure 6B) [181]. Another direction involves using nanotechnology-based imaging agents to monitor CAR-T cell distribution and activity, allowing for early detection and intervention if off-tumor effects occur [182]. Several strategies utilizing NPs have been proposed [182]. These strategies involve ex vivo preloading of NPs onto CAR-T cells. For instance, in mouse models, the biodistribution of CAR-T cells has been monitored by loading radiolabeled or contrast-agent NPs into CAR-T cells before infusion [183]. Light-induced release offers a novel approach to controlling CAR-T cell activation. A light-inducible nuclear translocation and dimerization (LINTAD) system allows precise control over CAR-T cell activation by using light to induce nuclear translocation and dimerization, thereby regulating gene expression in a highly specific manner [184]. These approaches allow for precise tracking and improved targeting of CAR-T cells, potentially enhancing their therapeutic efficacy [185].

## 6. Conclusions

Nanotechnology holds transformative potential for advancing CAR-T cell therapy by significantly enhancing multiple facets of the treatment process. It facilitates the precise construction of CARs, ensures efficient T cell transfection, boosts CAR-T cell expansion, allows for precise modifications, and provides advanced monitoring capabilities. Through the functionalization and surface modification of NPs, researchers can optimize CAR-T cell efficacy and effectively navigate the physical barriers inherent in solid tumors. Furthermore, nanotechnology offers valuable insights into the therapy’s effectiveness and potential adverse effects via real-time monitoring, enabling timely and precise adjustments to treatment protocols. To conclude, the integration of nanotechnology with CAR-T cell therapy not only enhances its effectiveness and safety but also personalizes treatment, thereby minimizing the associated risks and improving patient outcomes. However, there are significant challenges associated with the use of nanocarriers for delivering large antibodies, particularly concerning safety and toxicity. One of the primary concerns is the potential toxicity of inorganic nanoparticles, such as gold nanoparticles, which have been shown to increase reactive oxygen species levels, which can lead to oxidative stress and subsequent cellular damage [186,187]. Prolonged exposure to these nanoparticles can exacerbate these effects, leading to adverse side effects [187]. Controlling nanoparticle size is crucial; if nanoparticles are too large, they may evade clearance and accumulate, leading to toxicity by crossing cellular toxic thresholds and causing unintended biological effects [188,189]. These safety concerns necessitate careful consideration of nanoparticle design, including size optimization and surface modifications, to reduce potential toxicity [190]. Moreover, thorough preclinical testing and monitoring of long-term exposure effects are essential to mitigate the risks associated with the use of inorganic nanoparticles in therapeutic applications [191,192]. In addition, developing stable formulations presents significant challenges related to sterility and stability. For instance, the delicate fragile nature of nucleic acid macromolecules, such as those used in CAR-T cell therapy delivery systems, complicates the formulation process [193], necessitating the careful optimization of cryopreservation agents, the selection of suitable primary containers, and the design of effective freezing profiles [194]. These factors are critical for preserving vector activity throughout purification and storage, and for preventing issues such as aggregation, proteolysis, and oxidation [195]. Ensuring sterility further demands adherence to GMP-compliant environments, rigorous microbial testing, and thorough validation of closed-system manufacturing, particularly for viral vectors [196]. Stability concerns during cryopreservation, particularly with agents like DMSO, can compromise cell viability and function if not handled properly [196]. Moreover, regular stability testing and maintaining ultra-low temperatures during storage and transport are crucial for preserving product potency. These interconnected challenges highlight the complexity of creating scalable and robust formulations for clinical use [196].

## Figures and Tables

**Figure 1 pharmaceutics-16-01228-f001:**
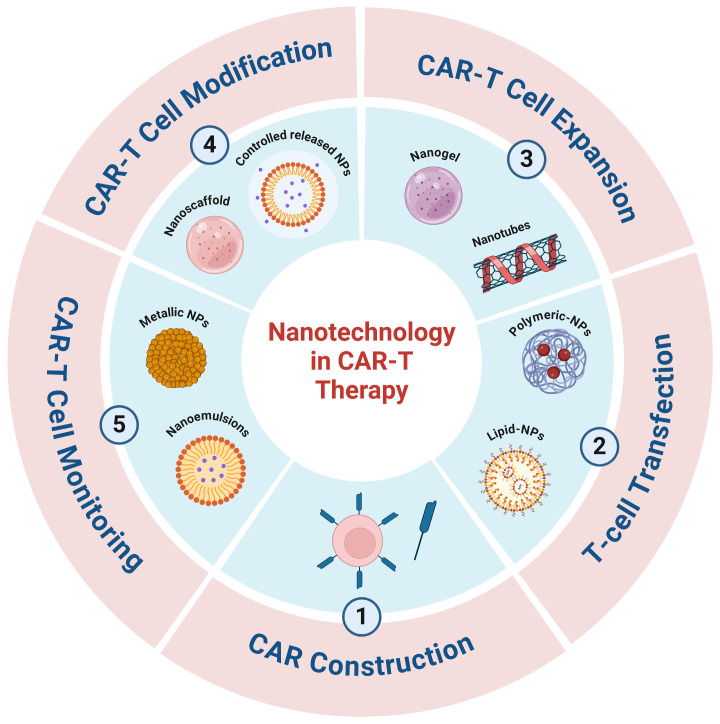
Nanoparticles enhancing CAR-T cell therapy. Nanoparticles (NPs) improve CAR-T cell therapy by facilitating efficient construction, transfection, expansion, and monitoring. NPs aid in gene delivery during transfection, enhance CAR-T cell expansion, and provide real-time monitoring capabilities, addressing challenges in treating cancer. Created with BioRender.com, accessed on 10 July 2024.

**Figure 2 pharmaceutics-16-01228-f002:**
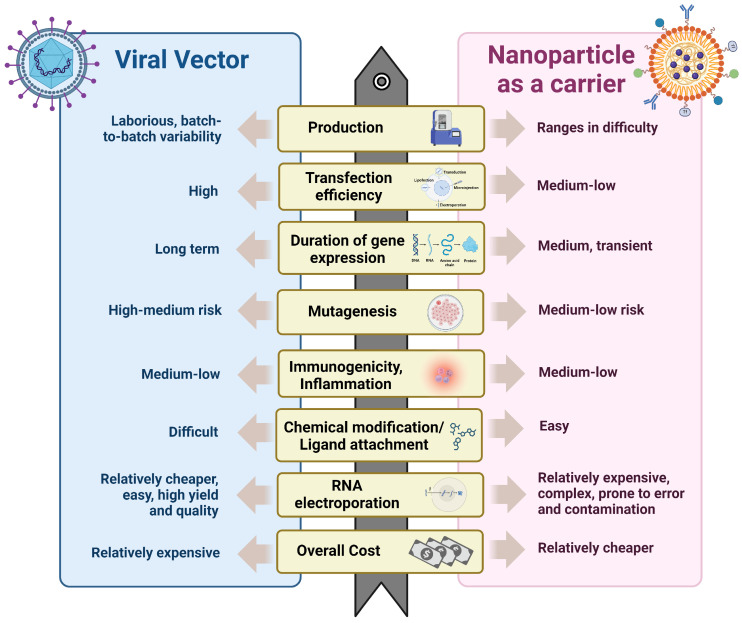
Illustration of the comparative benefits and challenges of using nanoparticles versus viral vectors in CAR-based therapies, highlighting the potential of nanotechnology to overcome current limitations in safety, efficacy, and cost. Created with BioRender.com, accessed on 10 July 2024.

**Figure 3 pharmaceutics-16-01228-f003:**
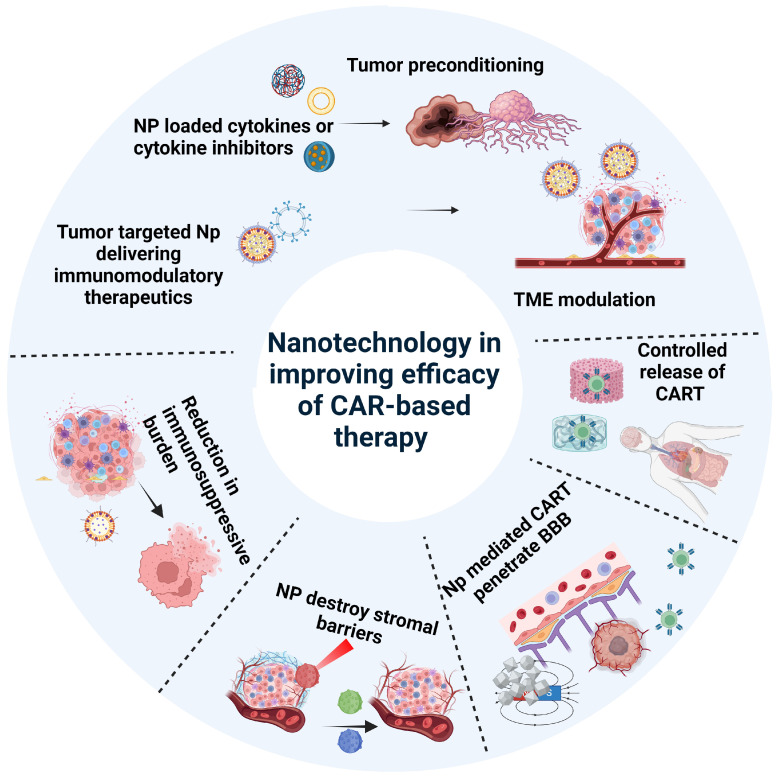
Nanotechnology serves as a versatile tool to enhance the precision and efficacy of CAR-T cell therapies. Specifically, nanotechnology can achieve controlled release of CAR-T cells, improve CAR-T cell penetration, destroy stromal cells, and reduce tumor burden. Additionally, nanotechnology can be used as a carrier to load immunomodulatory factors, cytokines, or associated inhibitors to modulate the tumor microenvironment (TME). Created with BioRender.com, accessed on 11 July 2024.

**Figure 4 pharmaceutics-16-01228-f004:**
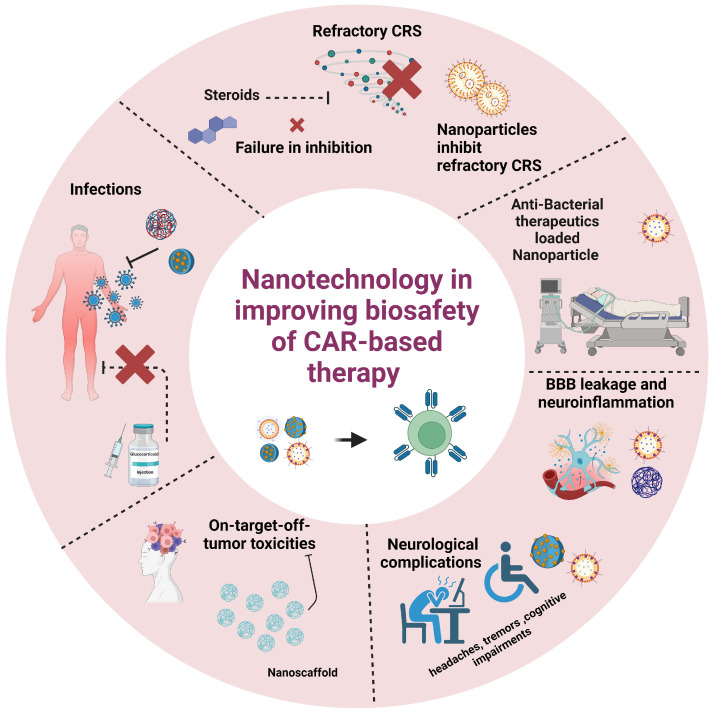
Nanoparticles (NPs) offer promising solutions to alleviate cytokine release syndrome (CRS) and immune effector cell-associated neurotoxicity syndrome (ICANS) and mitigate off-target effects and infections in CAR-T cell therapy. Created with BioRender.com, access 28 July 2024.

**Figure 6 pharmaceutics-16-01228-f006:**
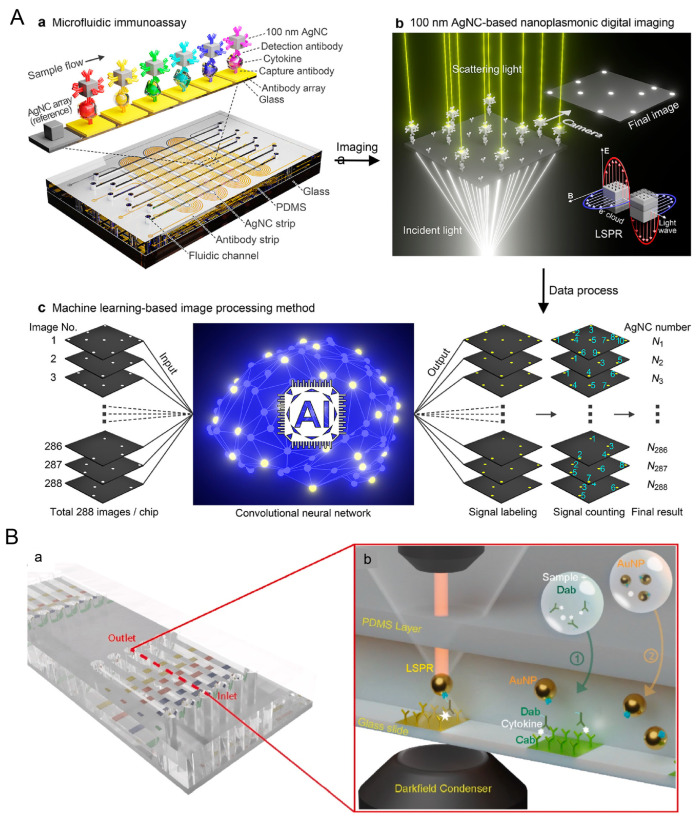
(**A**) A machine-learning-assisted microfluidic nanoplasmonic digital immunoassay has been developed for monitoring cytokine storms. © 2021 American Chemical Society. The immunoassay comprises three primary components: (**a**) the microfluidic immunoassay platform, (**b**) the nanoplasmonic digital imaging technology, and (**c**) the machine-learning-based image processing method [180]. (**B**) The detection of cytokines induced by CAR-T cells is facilitated using a digital nanoplasmonic microarray immunosensor. (**a**) A schematic illustrates the design of this microfluidic nanoplasmonic microarray immunosensor. (**b**) The principle behind this immunosensor involves cytokine molecules forming a sandwich structure on immobilized capture antibodies (CAbs) with detection antibodies (DAbs). These structures are then labeled with gold nanoparticles (AuNPs) that produce a strong nanoplasmonic scattering signal [181]. Copyright 2023 © 2023 Elsevier B.V.

**Table 1 pharmaceutics-16-01228-t001:** The CAR-T against brain tumors.

Therapy	Targeted Tumors	Targeted Antigen	Antigen expressions	Important Findings	Ref.
B7-H3 CAR-T cells	Pediatric solid tumors and brain tumors (osteosarcoma, medulloblastoma, and Ewing sarcoma)	B7-H3 (CD276)	Normally expressed in the liver, lung, bladder, testis, prostate, breast, placenta, and lymphoid organs, but highly expressed in high-grade gliomas and other brain tumors.	B7-H3 CAR-T cells demonstrated potential in treating specific solid tumors in vivo, with the effectiveness closely linked to the level of B7-H3 antigen expressed on the tumor cells, presenting a prospect for a therapy that can minimize damage to normal tissue while effectively targeting tumors.	[44]
CD133 (AC133-specific) CAR-T cells	Malignant glioblastoma multiforme (GBM)	CD133	Normally expressed in hematopoietic stem cells, endothelial progenitor cells, and neuronal stem cells; also expressed in glioma tumor-initiating cancer stem cells.	AC133-specific CAR-T cells demonstrated efficacy in specifically targeting and killing glioblastoma stem cells in vitro and in vivo. However, these interactions induce the T cell aging marker CD57, emphasizing the need to consider T cell differentiation markers like CD57 in therapeutic strategies.	[45]
CSPG4 CAR-T cells	Glioblastoma and neurospheres	Chondroitin sulfate proteoglycan 4 (CSPG4)	Normally expressed in chondroblasts, pericytes, and cardiomyocytes; also expressed uniformly in GBMs (67% high expression).	CSPG4-specific CAR-T cells successfully suppress tumor growth in vitro and in vivo, even in tumors with lower levels of CSPG4 expression caused by the increase in CSPG4 generated by TNFα from nearby microglia. Its dual expression pattern potentially reduces the risk of tumor escape, improving its efficacy in the treatment of GBM.	[46]
EGFRvIII-specific murine CAR (mCAR) and humanized EGFRvIII-directed T cells	Glioblastoma multiforme (GBM)	Epidermal growth factor receptor variant III (EGFRvIII)	Normally, the expression is restricted; expressed in the most common EGFR mutation in GBM (approximately 30% of GBMs).	EGFRvIII-specific mCAR-T cells demonstrated potent antitumor activity in brain tumors in vivo, relying on lymphodepletive conditioning for efficacy enhancement and neutralization of its activity using EGFRvIII soluble peptide to potentially improve therapy safety. While in glioblastoma xenogeneic models, humanized EGFRvIII-directed CAR-T cells controlled tumor growth, exhibiting specificity for EGFRvIII and validating functionality in vitro, paving the way for a phase 1 clinical study (NCT02209376) in patients with residual or recurrent glioblastoma.	[47,48]
EphA2 CAR-T cells	Glioblastoma (GBM)	Erythropoietin-producing hepatocellular carcinoma A2 (EphA2)	Normally expressed in epithelial tissue; also expressed uniformly in high-grade glioma with various levels.	EphA2 CAR-T cell therapy in glioblastoma demonstrates potent antitumor activity in vitro and in vivo, showing efficacy against EphA2-positive glioma cells and leading to regression of glioma xenografts in severe combined immunodeficiency (SCID) mice.	[49,50]
GD2-targeted CAR-T cell	Pediatric brain cancers (diffuse intrinsic pontine glioma (DIPG) and other diffuse midline gliomas (DMGs) with mutated histone H3 K27M (H3-K27M))	Glycoprotein disialoganglioside (GD2)	Normally expressed in the central nervous system, peripheral nerves, and skin melanocytes; also expressed uniformly in DIPGs; and low in high-grade gliomas.	H3-K27M-mutant glioma cells expressing high GD2 levels are effectively targeted by anti-GD2 CAR-T cells in orthotopic models, though neuroinflammation-induced hydrocephalus necessitates cautious clinical oversight.	[51]
HER2-CAR T-cell	Glioblastoma (GBM)	Human epidermal growth factor receptor 2 (HER2)	Normally expressed in the epithelial tissue, skin, and muscle; moderately expressed on GBM; and highly expressed on other solid tumors that metastasize to the brain.	HER2-specific T cells generated via gene transfer showed robust antitumor activity against HER2-positive GBM cells and their stem cells, inducing sustained regression of GBM xenografts in mice, highlighting their potential as a promising adoptive immunotherapy for GBM.	[52,53]
IL13-zetakine-redirected T cells	High-grade gliomas	IL–13 receptor α2 (IL13Rα2)	Normally expressed in testis. Also, expressed in the majority of GBM and other high-grade gliomas.	IL13-zetakine (+), cytolytic T lymphocytes (CTLs), and universal tricistronic transgene (UCAR-T cells) targeting HER2, IL13Rα2, and EphA2 can effectively recognize and eliminate glioma stem cells, overcoming antigenic heterogeneity and improving treatment outcomes in high-grade gliomas, while CD8(+) CTLs specifically target brain tumor stem cells, laying the foundation for curative immunotherapies.	[54,55,56]

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
