# Peer review of "Nanotechnology in Advancing Chimeric Antigen Receptor T Cell Therapy for Cancer Treatment"

_pharmaceutics, 2024, doi:10.3390/pharmaceutics16091228_

Round 1

Reviewer 1 Report

Comments and Suggestions for Authors

This Paper titled "Nanotechnology in Advancing Chimeric Antigen Receptor T-2 Cell Therapy for Cancer Treatment" is very interesting and has potential for publication but I may have some concerns.

Section 1.1  CAR-based Therapy in Cancer and the Role of Nanotechnology; the author may present some examples.

Also, it has some detailed information and is presented in a good way.

The author may focus on techniques for the fabrication of nanotechnology-enabled CAR therapies, characterization methods as well as challenges and perspectives in a more detailed way.

Comments on the Quality of English Language

English quality is overall ok. In some places, it can be revised for a better understanding of non-expert of the topics.

Author Response

Manuscript number and Title:

Reviewer 1

Dear Reviewer, Thank you for your efforts in reviewing the whole MS.

Comment 1

This Paper titled "Nanotechnology in Advancing Chimeric Antigen Receptor T-2 Cell Therapy for Cancer Treatment" is very interesting and has potential for publication but I may have some concerns.

Section 1.1  CAR-based Therapy in Cancer and the Role of Nanotechnology; the author may present some examples.

Also, it has some detailed information and is presented in a good way.

Response

Thank you for your valuable suggestions. We have incorporated some examples as per your advice, which can be found in Section 1, lines 65-108.The section now includes more detailed information, and we believe it is presented in a clearer and more structured manner. We greatly appreciate your opinion and have refined the details and layout accordingly. The following information has been included in the revised manuscript.

Please check the new information here. Line 70-94 in MS: please open track mode and choose no-markup

Bionanotechnology also allows for the control of side effects of CAR therapy large mo-lecular weight antibodies like tocilizumab is significant in the minimizing the side ef-fects of CAR therapy but remain challenges in delivery due to their size and complexi-ty. Strategies such as the use of hyaluronate-gold nanoparticles, as seen in tocilizumab formulations, and infliximab-modified gold nanorods, have shown promise.[16, 17] Another promising strategy involves the use of antibody-mimicking peptides, which are around 50 amino acids in length and can be engineered to replicate the binding specificity and affinity of the original antibodies while being significantly smaller in size.[18, 19] Notably, four phage mimics YHTTDKLFYMMR, YSAYEFEYILSS, KTMSAEEFDNWL, and LTSHTYRSQADT) have been shown to mimic the activity of tocilizumab.[20] Furthermore, utilizing stimuli-responsive nanovehicles that release their cargo in response to specific environmental triggers, such as pH changes or the presence of certain enzymes in the tumor microenvironment, can improve the circula-tion time of the antibodies in the bloodstream.[21] Nanodynamic therapies (NDTs) can activate anti-tumor immunity, thereby enhancing the efficacy of immunotherapy.[22] A notable example is photodynamic therapy, which utilizes nanoparticles to enhance the anti-tumor activity of CAR T cells. [23] Moreover, the potential for nanotechnology integrated with multitarget molecular therapies opens new avenues.[24, 25] CARs are engineered to redirect T lymphocytes and other immune cells towards specific antigens. [26] However, tumor-associated macrophages (TAMs), myeloid-derived suppressor cells (MDSCs), and cancer-associated fibroblasts (CAFs) can impair CAR-T cell effica-cy by producing inhibitory substances that affect the therapy's effectiveness. [27, 28] To address the challenges in CAR-T cell therapy, nanoparticles with multimolecular targets can be engineered to target those cells within the tumor microenvironment.[29]. [30] For instance, combining all-trans retinoic acid (ATRA) with CAR-T cell therapy can mitigate the suppressive effects of myeloid-derived suppressor cells (MDSCs).[31]. Besides, the decrease of TGF-b and IL10 using nanoparticle also benefits the efficacy of CAR-T. [32]The combination of CAR-T with chemotherapy delivered by np is able to inhibit the immunosuppressive cells including T reg and MDSC. [33, 34]The delivery of chemotherapy with nanoparticle can also reduce the burden of tumors and increase the damage associated with pattern release, which further potentially affects the ef-fects of CAR-T [33, 35].

Comment 2

The author may focus on techniques for the fabrication of nanotechnology-enabled CAR therapies, characterization methods as well as challenges and perspectives in a more detailed way.

Response

Thank you for your suggestion. We have provided detailed information on techniques such as electroporation, EP ligand modification, and others in Section 1, lines 161-169, lines 337-342, and lines 361-369. Additionally, we have incorporated the challenges and perspectives as recommended in the Conclusion section, acknowledging that these challenges persist and offering insights into future directions. We appreciate your thoughts and believe these additions have strengthened the manuscript. The following information has been included in the revised manuscript

Reviewer 2 Report

Comments and Suggestions for Authors

In their current work, the authors have systematically summarized the role and implications of nanotechnology in developing effective and safer CAR-T therapies against neoplasms.  The review is well written and needs minor additions to the following:

1.       Please include additional references/studies on ex-vivo transfection of RNA complexed nanoparticles into T-cells generating CARs and their benefits over electroporation.

2.       Please summarize similar technologies involving lipid nanoparticles targeting antigen presenting cells causing T-cell induction in preclinical in-vivo models.

3.       What are the technological challenges in developing formulations for ex-vivo T-cell transduction with respect to their sterility and stability considerations. 

Please include these additional details in the review for its completion to readers.  Thank you.

Author Response

In their current work, the authors have systematically summarized the role and implications of nanotechnology in developing effective and safer CAR-T therapies against neoplasms.  The review is well written and needs minor additions to the following:

Please include additional references/studies on ex-vivo transfection of RNA complexed nanoparticles into T-cells generating CARs and their benefits over electroporation.

Thank you for your valuable suggestions. We have incorporated some examples based on your advice, which are now available in line 143-161. The section has been expanded to include more detailed information, and we believe it is now presented in a clearer and more structured manner. We greatly appreciate your feedback.

Newly added information:

Viral vectors require a complex and costly manufacturing process, necessitating stringent safety protocols to avoid contamination [63]. Additionally, they have a limited packaging capacity, which restricts the size of CAR constructs [61]. These factors underscore the need for continued research to enhance the safety and efficacy of viral vector-based therapies. To address these challenges, non-viral physical methods have garnered significant attention. Electroporation, for example, has shown great promise by creating temporary pores in the cell membrane through the application of electrical fields, facilitating more efficient gene transfer [65, 66]. While microinjection is less frequently used for large-scale transduction due to its single-cell approach, it retains value for precision in immunotherapy applications [67]. Sonication, a newer technique, leverages ultrasonic waves to increase membrane permeability and aid gene uptake [68]. These physical methods not only enhance transfection efficiency but also reduce dependence on viral vectors [69]. However, a significant challenge remains, as these techniques may cause cell damage, presenting a major hurdle in their widespread application. Compare to viral vector and physical methods, nanotechnology offers a promising alternative for advancing CAR-based therapy[73].  Importantly, since nanomaterials are easier to manufacture, store, and transport, nanotechnology can significantly reduce the cost of CAR-based therapy.  In addition, NPs can serve as versatile delivery platforms for CAR genes and other therapeutic agents

Reference:

  1. Akinc, A., et al., The Onpattro story and the clinical translation of nanomedicines containing nucleic acid-based drugs. Nature nanotechnology, 2019. 14(12): p. 1084-1087.
  2. Billingsley, M.M., et al., Ionizable lipid nanoparticle-mediated mRNA delivery for human CAR T cell engineering. Nano letters, 2020. 20(3): p. 1578-1589.
  3. Filley, A.C., M. Henriquez, and M. Dey, CART immunotherapy: development, success, and translation to malignant gliomas and other solid tumors. Frontiers in oncology, 2018. 8: p. 409326.
  4. Akhavan, D., et al., CAR T cells for brain tumors: Lessons learned and road ahead. Immunological reviews, 2019. 290(1): p. 60-84.
  5. Huang, Z., et al., CAR T cells: engineered immune cells to treat brain cancers and beyond. Molecular cancer, 2023. 22(1): p. 22.
  6. Renner, K., et al., Metabolic Hallmarks of Tumor and Immune Cells in the Tumor Microenvironment. Front Immunol, 2017. 8: p. 248.
  7. Rodriguez-Garcia, A., et al., CAR-T cells hit the tumor microenvironment: strategies to overcome tumor escape. Frontiers in immunology, 2020. 11: p. 548227.

Please summarize similar technologies involving lipid nanoparticles targeting antigen presenting cells causing T-cell induction in preclinical in-vivo models.

Thank you, we increase the below content per your suggestion

Lipid nanoparticles (LNPs) are being used more frequently to target antigen-presenting cells (APCs) in order to stimulate T-cell responses in preclinical in vivo models.[2] Modified cationic lipids have been shown to trigger strong CD8+ and CD4+ T-cell responses. [3]And,PEGylated nanoparticles offer a biocompatible system for gene delivery, improving both the circulation time and stability of vaccines in vivo[4]. PEG-modified nanoparticles are good examples in providing a biocompatible platform for gene transfer. This  NP enhanced the stability of vaccines in vivo [5]. In addition, LNP can encapsulate antigens and deliver tumor antigen directly to APCs, such as dendritic cells (DCs).[6]  For instance, empty LNP (eLNP) induced maturation of DCs. [7]Besides, The administration of eLNP resulted in the upregulation of CD40 and the induction of cytokine production, exhibiting an age-dependent response [7]. Due to their lipid composition, dendritic cells (DCs) preferentially uptake LNPs.[8, 9] Furthermore, surface modifications of these nanoparticles can enhance their uptake by DCs, leading to improved antigen presentation and more effective T-cell activation.[10]

LNPs are also used to deliver mRNA encoding tumor antigens or immune-modulatory proteins to APCs.[11]).  Besides, the incorporation of other ingredient such as C-24 alkyl phytosterols can improve the gene transfection efficacy. [11] Rurik et al. [125] found coupling a CD5 antibody to an LNP  and form CD5/LNP can be used to deliver CAR mRNA  targeting fibroblast-activating protein (FAP) can minimize the  cardiac injury.[12] The LNP vector successfully targeted splenic T cells in mice with high CD5 expression, recognizing cardiac fibroblasts, ultimately reducing fibrosis and improving cardiac function[12].Furthermore, combining antigens with immune-stimulatory adjuvants in LNPs can further enhance T-cell activation.[13] Adjuvants such as Toll-like receptor (TLR) agonists or other immune modulators can be incorporated into LNPs to amplify the overall immune response.[14]  For instance, MHC class I antigenic-peptide ligand encapsulated in a LNP resulted in increased T cell expansion in vivo[14]

  1. Chuang, S.T., et al., Nanotechnology-enabled immunoengineering approaches to advance therapeutic applications. Nano Convergence, 2022. 9(1): p. 19.
  2. Liang, F., et al., Efficient targeting and activation of antigen-presenting cells in vivo after modified mRNA vaccine administration in rhesus macaques. Molecular Therapy, 2017. 25(12): p. 2635-2647.
  3. Liu, G., et al., Nanotechnology-empowered vaccine delivery for enhancing CD8+ T cells-mediated cellular immunity. Advanced drug delivery reviews, 2021. 176: p. 113889.
  4. Zhu, D., et al., Co-delivery of antigen and dual agonists by programmed mannose-targeted cationic lipid-hybrid polymersomes for enhanced vaccination. Biomaterials, 2019. 206: p. 25-40.
  5. Sekiya, T., et al., PEGylation of a TLR2-agonist-based vaccine delivery system improves antigen trafficking and the magnitude of ensuing antibody and CD8+ T cell responses. Biomaterials, 2017. 137: p. 61-72.
  6. Sasaki, K., et al., mRNA-loaded lipid nanoparticles targeting dendritic cells for cancer immunotherapy. Pharmaceutics, 2022. 14(8): p. 1572.
  7. Connors, J., et al., Lipid nanoparticles (LNP) induce activation and maturation of antigen presenting cells in young and aged individuals. Communications biology, 2023. 6(1): p. 188.
  8. Basha, G., et al., Influence of cationic lipid composition on gene silencing properties of lipid nanoparticle formulations of siRNA in antigen-presenting cells. Molecular Therapy, 2011. 19(12): p. 2186-2200.
  9. Firdessa-Fite, R. and R.J. Creusot, Nanoparticles versus dendritic cells as vehicles to deliver mRNA encoding multiple epitopes for immunotherapy. Molecular Therapy-Methods & Clinical Development, 2020. 16: p. 50-62.
  10. Katakowski, J.A., et al., Delivery of siRNAs to dendritic cells using DEC205-targeted lipid nanoparticles to inhibit immune responses. Molecular Therapy, 2016. 24(1): p. 146-155.
  11. Patel, S., et al., Naturally-occurring cholesterol analogues in lipid nanoparticles induce polymorphic shape and enhance intracellular delivery of mRNA. Nature communications, 2020. 11(1): p. 983.
  12. Rurik, J.G., et al., CAR T cells produced in vivo to treat cardiac injury. Science, 2022. 375(6576): p. 91-96.
  13. Swaminathan, G., et al., A novel lipid nanoparticle adjuvant significantly enhances B cell and T cell responses to sub-unit vaccine antigens. Vaccine, 2016. 34(1): p. 110-119.
  14. Su, F.-Y., et al., In vivo mRNA delivery to virus-specific T cells by light-induced ligand exchange of MHC class I antigen-presenting nanoparticles. Science Advances, 2022. 8(8): p. eabm7950.

What are the technological challenges in developing formulations for ex-vivo T-cell transduction with respect to their sterility and stability considerations. 

Thank you for your suggestion, we add the below content per your suggestion

Developing stable formulations for ex-vivo T-cell transduction presents significant challenges related to sterility and stability. For instance, the delicate fragile nature of nucleic acid macromolecules, such as those used in CAR-T cell therapy delivery systems, complicates the formulation process [1], necessitating careful optimization of cryopreservation agents, selection of suitable primary containers, and design of effective freezing profiles [2]. These factors are critical for preserving vector activity throughout purification and storage, and for preventing issues such as aggregation, proteolysis, and oxidation [3]. Ensuring sterility further demands adherence to GMP-compliant environments, rigorous microbial testing, and thorough validation of closed-system manufacturing, particularly for viral vectors [4]. Stability concerns during cryopreservation, particularly with agents like DMSO, can compromise cell viability and function if not handled properly [4]. Moreover, regular stability testing and maintaining ultra-low temperatures during storage and transport are crucial for preserving product potency. These interconnected challenges highlight the complexity of creating scalable and robust formulations for clinical use [4].

Ref:

[1]        B. K. Muralidhara, R. Baid, S. M. Bishop, M. Huang, W. Wang, and S. Nema, “Critical considerations for developing nucleic acid macromolecule based drug products,” Drug Discov. Today, vol. 21, no. 3, pp. 430–444, Mar. 2016, doi: 10.1016/j.drudis.2015.11.012.

[2]        C. F. van der Walle, S. Godbert, G. Saito, and Z. Azhari, “Formulation Considerations for Autologous T Cell Drug Products,” Pharmaceutics, vol. 13, no. 8, p. 1317, Aug. 2021, doi: 10.3390/pharmaceutics13081317.

[3]        J. F. Wright, G. Qu, C. Tang, and J. M. Sommer, “Recombinant adeno-associated virus: formulation challenges and strategies for a gene therapy vector,” Curr. Opin. Drug Discov. Devel., vol. 6, no. 2, pp. 174–178, Mar. 2003.

[4]        C. Roddie, M. O’Reilly, J. Dias Alves Pinto, K. Vispute, and M. Lowdell, “Manufacturing chimeric antigen receptor T cells: issues and challenges,” Cytotherapy, vol. 21, no. 3, pp. 327–340, Mar. 2019, doi: 10.1016/j.jcyt.2018.11.009.

Please include these additional details in the review for its completion to readers.  Thank you.

Reviewer 3 Report

Comments and Suggestions for Authors

The review “Nanotechnology in Advancing Chimeric Antigen Receptor T Cell Therapy for Cancer Treatment” by Chen et al has been submitted for the publication to Pharmaceutics. The manuscript deals with several kinds of potential application of nanotechnology of CAR T Cell therapy for Cancer.

The treated issue is very interesting and the cited references are high-quality papers but the review can be published only after a major revision concerning the following points:

1. The structure of the review is unconventionally shifted toward the main theme without a general introduction that is completely absent and should be added. Successively a new paragraphing of the manuscript will be requested. According to this reviewer’s opinion this general introduction should be centred on the existing several different approaches to cancer therapeutics through nanotechnologies spanning from bionanotechnologies to nanodynamic therapies and also considering perspectively glimpsed extension of nanotechnology approach to multitarget molecular therapies. Several authoritative reviews at least 6-8 should be introduced, below you can find two suggestions:

https://doi.org/10.1016/j.mtbio.2022.100472

DOI: 10.1002/smsc.202400113     

2. Sub-paragraph 4.2, lines 291-300: Try to give further details on the imagined  rategies to the cited tocilizumab siltuximab, infliximab, etanercept and anakinra on nanovehicles. The cited drugs are antibodies characterized from very large MW. The sentence at lines 299-300 is too generic. One of the above-cited strategies could be to employ antibody-mimicking peptides.

3. Paragraph 5, lines 382-383; “CARs engineered to target small molecules can be redirected to various 382 tumor antigens via conjugated antibodies.” Better explain the above reported sentence.

Minor issue:

Ref. 80 seems to be not complete.

Author Response

Reviewer 2

The review “Nanotechnology in Advancing Chimeric Antigen Receptor T Cell Therapy for Cancer Treatment” by Chen et al has been submitted for the publication to Pharmaceutics. The manuscript deals with several kinds of potential application of nanotechnology of CAR T Cell therapy for Cancer.

Dear Reviewer, Thank you for your efforts in reviewing the whole MS.

The treated issue is very interesting and the cited references are high-quality papers but the review can be published only after a major revision concerning the following points:

Comment 1

The structure of the review is unconventionally shifted toward the main theme without a general introduction that is completely absent and should be added. Successively a new paragraphing of the manuscript will be requested. According to this reviewer’s opinion this general introduction should be centred on the existing several different approaches to cancer therapeutics through nanotechnologies spanning from bionanotechnologies to nanodynamic therapies and also considering perspectively glimpsed extension of nanotechnology approach to multitarget molecular therapies. Several authoritative reviews at least 6-8 should be introduced, below you can find two suggestions:

https://doi.org/10.1016/j.mtbio.2022.100472

DOI: 10.1002/smsc.202400113     

Response

Thank you for your thorough and constructive feedback on our manuscript. We apologize the incorrect layout in the manuscript, we provided the introduction, however, we wrongly labelled as 1.1.1-1.1.5 and now we corrected them. So, now, based on your useful suggestion, we firstly introduced the general introduction of cancer, please see line 30-34, then we bring the introduction of CAR therapy and how nanoparticles benefits CAR therapy  ( line 39-66), .In addition, we also increase other contents including bionanotechnologies, nanodynamic therapies (NDTs), multitarget molecular therapies per your request,we also incorporate the reference you mentioned (line 75-109). The following information has been included in the revised manuscript.

Bionanotechnology also allows for the control of side effects of CAR therapy large mo-lecular weight antibodies like tocilizumab is significant in the minimizing the side ef-fects of CAR therapy but remain challenges in delivery due to their size and complexi-ty. Strategies such as the use of hyaluronate-gold nanoparticles, as seen in tocilizumab formulations, and infliximab-modified gold nanorods, have shown promise.[16, 17] Another promising strategy involves the use of antibody-mimicking peptides, which are around 50 amino acids in length and can be engineered to replicate the binding specificity and affinity of the original antibodies while being significantly smaller in size.[18, 19] Notably, four phage mimics YHTTDKLFYMMR, YSAYEFEYILSS, KTMSAEEFDNWL, and LTSHTYRSQADT) have been shown to mimic the activity of tocilizumab.[20] Furthermore, utilizing stimuli-responsive nanovehicles that release their cargo in response to specific environmental triggers, such as pH changes or the presence of certain enzymes in the tumor microenvironment, can improve the circula-tion time of the antibodies in the bloodstream.[21]

Nanodynamic therapies (NDTs) can activate anti-tumor immunity, thereby enhancing the efficacy of immunotherapy.[22] A notable example is photodynamic therapy, which utilizes nanoparticles to enhance the anti-tumor activity of CAR T cells. [23] Moreover, the potential for nanotechnology integrated with multitarget molecular therapies opens new avenues.[24, 25] CARs are engineered to redirect T lymphocytes and other immune cells towards specific antigens. [26] However, tumor-associated macrophages (TAMs), myeloid-derived suppressor cells (MDSCs), and cancer-associated fibroblasts (CAFs) can impair CAR-T cell efficacy by producing inhibitory substances that affect the therapy's effectiveness. [27, 28] To address the challenges in CAR-T cell therapy, nanoparticles with multimolecular targets can be engineered to target those cells within the tumor microenvironment.[29]. [30] For instance, combin-ing all-trans retinoic acid (ATRA) with CAR-T cell therapy can mitigate the suppres-sive effects of myeloid-derived suppressor cells (MDSCs).[31]. Besides, the decrease of TGF-b and IL10 using nanoparticle also benefits the efficacy of CAR-T. [32]The combi-nation of CAR-T with chemotherapy delivered by np is able to inhibit the immunosup-pressive cells including T reg and MDSC. [33, 34]The delivery of chemotherapy with nanoparticle can also reduce the burden of tumors and increase the damage associated with pattern release, which further potentially affects the effects of CAR-T [33, 35].

Comment 2

Sub-paragraph 4.2, lines 291-300: Try to give further details on the imagined  rategies to the cited tocilizumab siltuximab, infliximab, etanercept and anakinra on nanovehicles. The cited drugs are antibodies characterized from very large MW. The sentence at lines 299-300 is too generic. One of the above-cited strategies could be to employ antibody-mimicking peptides.

Response

Thank you for your valuable feedback. We agree with your suggestions and have made the necessary improvements to the manuscript (Line 336-345). Specifically, we have added antibody-mimicking peptides based on your recommendations (Line 82-86). The following information has been included in the revised manuscript.

Tocilizumab, an antibody against IL6, has been approved for CRS treatment [137].  However, its use is often limited by issues such as cytopenias, resultant rheumatoid arthritis, and subsequent infections [138].  Using a nanocarrier loaded with tocilizumab may prevent premature release and avoid these potential risks. Other antagonists, such as siltuximab [139], infliximab [140], and etanercept [141], as well as the IL-1 antagonist anakinra [142], have also been discovered. Nanotechnology, which relies on nanomaterials to shield drugs, shows potential in providing sustained, controlled release and preventing drug degradation [143]. Additionally, modifications in nanomaterials can reduce untargeted toxicity [144]. Therefore, incorporating these anti-CRS agents with nanotechnology can be beneficial.

Comment 3

Paragraph 5, lines 382-383; “CARs engineered to target small molecules can be redirected to various 382 tumor antigens via conjugated antibodies.” Better explain the above reported sentence.

Response

Apologies for the previous incorrect expression. To clarify, we modified the contents,  Chimeric Antigen Receptors are engineered to redirect T lymphocytes and other immune cells towards specific tumor antigens. However, tumor-associated macrophages (TAMs), myeloid-derived suppressor cells (MDSCs), and cancer-associated fibroblasts (CAFs) can impair CAR-T cell efficacy by producing inhibitory substances that negatively impact the therapy’s effectiveness .To address these challenges, nanoparticles can be engineered to target these inhibitory cells within the tumor microenvironment. For instance, combining all-trans retinoic acid (ATRA) with CAR-T cell therapy can mitigate the suppressive effects of MDSCs. The following information has been included in the revised manuscript.

Minor issue:

Ref. 80 seems to be not complete.

Thank you, we now changed it

Round 2

Reviewer 3 Report

Comments and Suggestions for Authors

I have revised the manuscript in the form attached to your e-mail. I have noticed that Reference 18 and 25 are identical. I think that the authors should replace the actual ref 18 with the following one as stated in the revision https://doi.org/10.1016/j.mtbio.2022.100472. The manuscript, after this correction, can be accepted for the publication.